# Comparing the Effects of Encapsulated and Non-Encapsulated Propolis Extracts on Model Lipid Membranes and Lactic Bacteria, with Emphasis on the Synergistic Effects of Its Various Compounds

**DOI:** 10.3390/molecules28020712

**Published:** 2023-01-11

**Authors:** Luka Šturm, Ilja Gasan Osojnik Črnivec, Iztok Prislan, Nataša Poklar Ulrih

**Affiliations:** Biotechnical Faculty, University of Ljubljana, Jamnikarjeva 101, 1000 Ljubljana, Slovenia

**Keywords:** liposomes, lipid phase transition, chrysin, quercetin, (-)-epigallocatechin-3-gallate, trans-ferulic acid, caffeic acid, UV-oxidation

## Abstract

Propolis is a resinous compound made by bees with well-known biological activity. However, comparisons between encapsulated and non-encapsulated propolis are lacking. Therefore, the antibacterial activity, effect on the phase transition of lipids, and inhibition of UV-induced lipid oxidation of the two forms of propolis were compared. The results showed that non-encapsulated propolis produces quicker effects, thus being better suited when more immediate effects are required (e.g., antibacterial activity). In order to gain an in-depth introspective on these effects, we further studied the synergistic effect of propolis compounds on the integrity of lipid membranes. The knowledge of component synergism is important for the understanding of effective propolis pathways and for the perspective of modes of action of synergism between different polyphenols in various extracts. Thus, five representative molecules, all previously isolated from propolis (chrysin, quercetin, *trans*-ferulic acid, caffeic acid, (-)-epigallocatechin-3-gallate) were mixed, and their synergistic effects on lipid bilayers were investigated, mainly using DSC. The results showed that some compounds (quercetin, chrysin) exhibit synergism, whereas others (caffeic acid, *t*-ferulic acid) do not show any such effects. The results also showed that the synergistic effects of mixtures composed from several different compounds are extremely complex to study, and that their prediction requires further modeling approaches.

## 1. Introduction

Propolis is a sticky, lipophilic substance collected by bees from different plant sources, thus varying in composition among different propolis types (e.g., poplar propolis, Brazilian green propolis, etc.). Poplar propolis, the most commercially available propolis type, is generally composed of 50% plant balsam and resin, 30% beeswax, 10% essential and aromatic oils, 5% pollen, and 5% other various organic and inorganic compounds [1,2]. Over 850 compounds have been identified in propolis samples around the world, including phenolic acids, flavonoids, and terpenes [3], to which most of the biological activities of propolis have been attributed [4,5]. Despite the good antioxidant activity of propolis, as well as its antimicrobial and antitumor activity, its major limitation for use lies in its lipophilic nature, which is reflected in its poor water solubility [5]. Therefore, in order to use it as a food additive or for various medicinal applications, it is desirable to either remove its lipophilic compounds [6] or otherwise circumvent its poor water solubility. One of the most popular and industrially effective methods for such application is the encapsulation of propolis in various organic wall materials [7,8,9,10]. All kinds of encapsulated propolis powders are already available on the market, yet despite the very well-known effects of propolis, there are relatively few studies that address the direct effects of propolis on lipid bilayers [11], as scientists usually study its effects on bacteria [12,13], viruses [14], fungi [15,16], protists [17], or cancer cells [18]. The comparison between encapsulated and non-encapsulated propolis formulations is also rather scarce; the results obtained vary considerably from study to study, with some demonstrating the superior effect of extracts [12,19] and other encapsulated forms of propolis [20,21]. The best information on the effects of propolis on membranes can be found in the studies that have examined the effects of individual compounds such as catechins [22], phenolic acids [23], flavonoids [24,25,26,27], and terpenes [28,29,30] present in various propolis extracts. As most of the compounds investigated alter the rigidity/fluidity of the membranes, the knowledge of their synergistic effects as well as the effect of propolis on those membranes, in combination with other drugs, might be used in the treatment of, e.g., certain infections. Therefore, understanding the effects of these compounds as well as propolis products on lipid bilayers could prove crucial for potential new medicinal applications [11,31]. However, it is difficult to study the effects of individual compounds on the original cell membranes, so model lipid membranes are used instead. For the most basic studies, simple membranes consisting of only one or a few lipids are used, since they enable easier analysis of the obtained results [32]. In this way, it is possible at least to estimate how a specific compound, or a mixture of compounds, will affect the more complex lipid bilayers. Alas, since propolis is a mixture of many compounds, usually at least 80–100 per sample [33], the compounds can also interact among themselves, and consequently, their effects could be enhanced (synergism) or weakened (antagonism) [34,35,36,37]. Therefore, it is also important to study the possible synergistic effects of the different compounds in propolis. Understanding how the synergism of compounds in propolis affects lipid membranes is important not only for understanding how propolis extracts or their encapsulated powders work or how best to use them, but also from the perspective of how synergism between different polyphenols in different extracts might work in general and how we can use this knowledge for medicinal, food, cosmetic, or other purposes. In this article, we compared the effects of propolis ethanol extract (PP) and its encapsulated propolis powders in gum Arabic (LIO) [10] on the 1,2-dipalmitoyl-*sn*-glycero-3-phosphatidylcholine (DPPC) liposomal bilayers, as well as their antimicrobial and antioxidant effects. We also studied the individual and combined (synergistic) effects of the chosen propolis compounds on the model lipid membranes composed from either DPPC or phospholipon 90G (PL-90G) phospholipids.

## 2. Results and Discussion

### 2.1. Effects of Encapsulated and Non-Encapsulated Propolis on Lipid Membrane Integrity and Lactic Bacteria

Encapsulation is a technique that is useful for preparing various compounds/mixtures to increase their stability and extend their shelf life, mask their odor and taste, and allow for the solubilization of otherwise insoluble compounds as well as the gradual release of the encapsulated compound/mixture over a longer period of time/at the desired time, etc. [38,39]. However, in addition to all these advantages, encapsulation often hinders certain effects of the encapsulated compounds, e.g., reducing their cytotoxicity, antibacterial activity, and immediate antioxidant activity [19,40,41]. Therefore, encapsulation of certain compounds should be finely tuned to specific applications. To determine the application potential of encapsulated propolis, a comparison was made between encapsulated propolis powder (LIO) and the ethanol propolis extract (PP). First, their effect on the phase transition temperature of DPPC lipids was compared. As shown in Figure 1, PP has a stronger effect on the *T*_m_ than LIO, even when a lower ratio of propolis in the case of PP (*w*_PP_:*w*_DPPC_ = 1:20) is compared to a higher ratio of propolis in the case of LIO (*w*_LIO_:*w*_DPPC_ = 1:10). The results were expected, since not all compounds from LIO were released at once, and the hydrophobic molecules were most likely not released at all. This is also consistent with some other studies, in which it was shown that propolis extract was more cytotoxic to cancer cells than encapsulated powders, implying that the extract has a stronger effect on membranes than encapsulated powders [40].

Similar results were obtained in the antimicrobial activity tests (Table 1), where five bacterial strains were selected. Several strains of gram-positive bacteria were mainly selected, as they are generally more sensitive to propolis than gram-negative bacteria [13,42,43], and because they are considered to be probiotics [44,45]. Meanwhile, *E. coli* was also chosen to include a representative of gram-negative bacteria. It was found that LIO did not affect the growth of the bacterial strains at all, regardless of the concentration used (0.02, 0.2, 2, 5 and 10 mg/mL; all in water; results not shown), whereas even the lower concentrations of PP already affected the growth of *Lacticaseibacillus rhamnosus*, which proved to be the most sensitive to PP among the selected strains. In our previous study, the corresponding *L. rhamnosus* GG strain (ATCC 53103) also showed the lowest resistance to other inhibitory environmental factors among the tested lactic acid bacteria [46]. On the other hand, the growth of *E. coli* was not affected by any concentration of either PP or LIO (not even by 100% ethanol (EtOH)), and all other strains were affected by the highest concentrations of PP (10 and 200 mg/mL). Among the gram-positive strains, *Lactiplantibacillus plantarum* was the most resistant, which is again in agreement with our study in [46] and other previous studies [43,47]. It is also important to note that it was impossible to prepare the solution with higher concentrations of propolis than 10 mg/mL in the case of LIO, as the solubility of encapsulated powders played a large role in the preparation of such solutions. In our case, though LIO proved to be inactive against bacteria at the applied concentrations, the effect of encapsulated propolis on bacterial activity differs among other authors. Although many agree that encapsulated propolis has lower antibacterial activity than propolis extract [12,16,19,40,48,49], others have observed increased inhibitory effects of encapsulated propolis in comparison to their extracts [20,21,50,51]. The best evaluation was probably given by Jansen-Alves et al. [49], who found that the activity of encapsulated propolis depended on the percentage of wall material used for encapsulation—in other words, antibacterial activity was dependent on the mass ratio of propolis:wall material. Thus, encapsulated propolis inhibited bacterial growth in ratios that favored propolis, but it did not in ratios that favored wall material. It is important to note that different wall materials were used in the above studies, which have different effects on the encapsulation process and subsequent release of propolis. Furthermore, certain wall materials might even act as synergists in bacterial inhibition, as was the case with silver nanoparticles [50]. However, it is generally accepted that in the case of encapsulation of propolis in complex protein/polysaccharide wall materials, the antibacterial activity of such encapsulants is lower compared to propolis extract [48,49]. Since the ratio of propolis:wall material in the case of our LIO is in favor of wall material (1:4), it should not be surprising that LIO did not inhibit bacterial growth. Thus, propolis encapsulated in polysaccharide wall materials could be used when the decrease in microbial count is disfavored (e.g., at the point of ingestion), but it is required later on at the targeted location (e.g., in the colon).

In addition to comparing the membrane action and antibacterial activity of PP and LIO, we also wanted to compare their antioxidant activity in the case of UV radiation. With the experiment, we wanted to test whether both of them can be used in creams and lotions, since propolis has been used mainly in various ointments in the past [33]. To determine the antioxidant activity of PP and LIO, a TBARS test was performed. Initially, the tests were performed for PP, because it was assumed that the antioxidant effect of PP would be more pronounced based on the results of DSC and antibacterial tests. The results showed that both ratios of PP completely inhibited the oxidation of unsaturated lipids even after 24 h (*w*_PP_:*w*_PL-90G_ = 1:10 - results not shown; *w*_PP_:*w*_PL-90G_ = 1:100—Figure 2). Therefore, only the *w*_PP_:*w*_PL-90G_ = 1:100 ratio was performed for LIO, as it was expected that more significant differences would be obtained at this ratio. The results in Figure 2 show that although LIO inhibits lipids oxidation, its antioxidant effect is much less pronounced than in the case of PP. After 3 h, LIO could still completely inhibit the oxidation of lipids, as no statistical differences were found, but after 6 h, the difference in antioxidant activity was statistically significant. The greatest difference was observed after 24 h, when considerable oxidation of lipids containing LIO was detected. Based on the previous results (Figure 1 and Table 1), this was expected, although some researchers observed longer-lasting effects for encapsulated compounds compared to their free forms due to the slow and gradual release from the wall material [41]. In the case of LIO, some hydrophilic compounds were likely not released from the powders during the assay, and hydrophobic compounds were likely not released in the aqueous solution at all. This is also confirmed by our previously published results [10], where good antioxidant activity was shown for LIO in oils, which means that most bioactive compounds were released from propolis in hydrophobic media.

### 2.2. Synergistic Effects of Compounds from Propolis on the Gel-to-Liquid Crystalline Phase Transition of DPPC Lipids

Synergism of compounds in propolis was studied using five model compounds, namely: chrysin (CHRY), quercetin (QUER), EGCG, *trans*-ferulic acid (*t*-FER), and caffeic acid (CAFF) (Figure 3). All five were selected based on their demonstrated biological activity [52,53,54,55,56] and their occurrence in propolis samples [4,13,57]. Their effects on the lipid bilayer and possible synergism were studied on liposomes prepared from DPPC lipids by observing changes in the temperature of gel-to-liquid crystalline phase transition (*T*_m_) in DSC thermograms. The DPPC lipids were chosen, since their phase transition at 41 °C can be easily followed by the majority of experimental techniques. By using the liposomes prepared solely from DPPC lipids, the effect of single compounds or their mixture can be determined directly from the thermograms. At first, we started to examine the effect of pure compounds on *T*_m_. Different amounts of compounds were added to DPPC lipids, so that the final molar ratios were as follows: *n*_compound_:*n*_DPPC_ = 1:1, 1:5, and 1:10 for all compounds except QUER, and for QUER, a ratio of *n*_quercetin_:*n*_DPPC_ = 1:2 was used instead of *n*_quercetin_:*n*_DPPC_ = 1:1. This was done because the ratio of *n*_quercetin_:*n*_DPPC_ = 1:1 caused problems with the solubility of QUER, and at the same time, it led to a very strong aggregation of liposomes. Further, *n*_compound_:*n*_DPPC_ = 1:10 ratios were not measured for *t*-FER and CAFF, as even their *n*_compound_:*n*_DPPC_ = 1:1 and 1:5 ratios had virtually no effect on the *T*_m_. It should be noted that the high compound(s):lipid ratios were selected only in order to amplify the effects of selected compounds and to amplify the synergistic effects of these compounds on the lipid membranes. Concentrations that high would otherwise likely prove fatal for living cells in cell-based assays [58,59]. In addition, the final compound:compound ratios used in the studies of synergistic effects were not based on the actual ratios found in propolis, but were chosen only to demonstrate how the effects of certain compounds on the lipid bilayers are amplified when mixed with certain other compounds.

Figure 4 and Table 2 show the different effects of each chosen compound on *T*_m_ and the change in the enthalpy of the main thermally induced phase transition of DPPC*,* Δ*H,* for a given molar compound:DPPC ratio. As can be seen from the thermograms (Figure 4), QUER has the strongest effect on *T*_m_, followed by EGCG. CHRY showed a detectable effect, whereas *t*-FER and CAFF showed virtually no effect at all. Though QUER lowered *T*_m_ significantly at the highest concentration for even more than 5 °C, CHRY lowered *T*_m_ for approximately 1 °C even at the highest concentration. In the case of EGCG, two *T*_m_ peaks appear, instead of one, making the change in *T*_m_ more complex to analyze. This phenomenon was described previously [60], and thus it will not be further commented here. As for *t*-FER and CAFF, they seem not to affect the *T*_m_ of DPPC lipids’ main phase transition at least at the chosen pH. In the case of the change in the enthalpy of the main thermally induced phase transition of DPPC, Δ*H*_1+2_ of EGCG and CHRY increases with the increasing concentration of the compound, whereas in the case of *t*-FER and CAFF, it remains statistically unchanged, even when compared to pure DPPC. In the case of QUER, however, Δ*H*_1+2_ increases at first (1:5–QUER:DPPC ratio), but subsequently considerably decreases at the highest QUER concentration. This is most probably linked to the higher binding affinity and effect of QUER on the DPPC lipid membranes in comparison to other compounds chosen [26,61,62,63]. The *T*_m_ results for CAFF and *t*-FER are similar to the ones published previously. The slight differences observed are likely due to a different insertion of compounds into the membranes, since MLVs were used in our study, as opposed to the ULVs used by Ota et al. [23]. Though no differences were observed in *T*_m_ for CHRY [64] and EGCG [22,60] compared to other studies, there seems to be a major difference in how QUER affected the phase transition of DPPC [62,63]. The results can be explained by the fact that though others incorporated QUER into the DPPC lipid membranes during the creation of liposomes [62,63], in our study, QUER was added to the already formed liposomes right before the DSC experiments. This can be further supported by comparing the first and second heating scans of QUER (Figure 5), in which major differences between both scans can be observed. This is not only surprising, but also completely opposite compared to the first and second heating scans of other compounds, which were more or less super imposable (results not shown), as is usually expected [22,23,62,63,64]. The heating above the phase transition of DPPC lipids, in our case to 70 °C, seems to help QUER incorporate into the DPPC lipid membranes in the liquid crystalline state fully, as it was similarly observed in other studies [62,63]. This indicates that QUER interacts differently with lipids in gel and liquid crystalline form, as opposed to other chosen compounds. Consequently, the results not only show that the different compounds and their ratios affect the phase transition of the membranes differently, but also that the effects can differ based on how/when the compounds are added/incorporated into the membranes. In any case, the results clearly show that some compounds affect the phase transition of the DPPC lipids significantly (EGCG, CHRY, QUER), whereas others do not affect it at all (CAFF, *t*-FER).

For the study of synergistic effects, a mixture of all five compounds was prepared in a *n*_CHRY_:*n*_CAFF_:*n_t_*_-FER_:*n*_QUER_:*n*_EGCG_:*n*_DPPC_ = 2:2:2:1:2:10 ratio, corresponding to a *n_Mix_*:*n*_DPPC_ = 9:10 ratio. The results were compared to *n*_compound_:*n*_DPPC_ = 1:1 (or *n*_QUER_:*n*_DPPC_ 1:2 in the case of QUER) ratios of the pure compounds. As shown in Table 2 and Figure 6, the effect of the *Mix* on the *T*_m_ was significantly stronger than that of EGCG, CHRY, *t*-FER or CAFF. Nevertheless, the effect of the *Mix* on *T*_m_ was comparable to that of QUER, despite the ratio of pure QUER (*n*_QUER_:*n*_DPPC_ = 1:2) being lower than the ratio of the *Mix* (*n_Mix_*:*n*_DPPC_ = 9:10) or all other individual compounds (*n*_compound_:*n*_DPPC_ = 1:1). The same was observed when comparing the changes in enthalpy of the main thermally induced phase transition of DPPC caused by *Mix* and the chosen compounds. This shows that QUER impacts *T*_m_ of DPPC the strongest. However, since QUER represents only one of the components in *Mix, t*-FER and CAFF showed no previous effect on DPPC at all, and the shift in *T*_m_ of *Mix* and QUER is similar, this strongly indicates that two or more compounds in *Mix* most likely exhibit synergistic effect.

In order to demonstrate the extent of synergistic effects of the compounds, mixtures containing either CHRY or CAFF and QUER were prepared. These three compounds were chosen because they are among the most typical components of propolis [4,13,65,66,67], as their synergism with other compounds has been indicated from previous studies [34,35,36,68,69], and also because we observed their individual effects on the *T*_m_ to be quite different (Figure 4). The obtained results are gathered in Figure 7 and Figure 8, as well as Table 3. As observed in Figure 7, the synergism between QUER and CHRY exists, as neither pure QUER nor CHRY (in a *n*_CHRY/QUER_:*n*_DPPC_ = 3:10 ratio) had as pronounced effect as the QUER/CHRY mixture (*n*_QUER_:*n*_CHRY_:*n*_DPPC_ = 1:2:10 ratio → *n*_QUER+CHRY_:*n*_DPPC_ = 3:10 ratio). Although pure CHRY had a much lower effect than pure QUER, the shift in *T*_m_ (*ΔT*_m_) caused by the mixture of two compounds was 4.2 °C, whereas the shifts caused by pure components were 0.6 °C for CHRY and 2.9 °C for QUER in comparable molar ratios. This was not entirely unexpected, however, since a similar effect was reported previously [34]. The observations in Figure 7 are also further supported by the changes in enthalpy shown in Table 3, where the enthalpy of the main thermally induced phase transition of DPPC with the added CHRY/QUER mixture is 19.8 kJ/mol, whereas the enthalpy of the thermally induced main phase transition of DPPC with added QUER in a comparable molar ratio (*n*_QUER_:*n*_DPPC_ = 3:10) is 24.3 kJ/mol, and with added CHRY in a comparable molar ratio (*n*_CHRY_:*n*_DPPC_ = 3:10), it is 31.5 kJ/mol. On the other hand, no synergism was observed in the case of QUER and CAFF (Figure 8), as the peak of the QUER/CAFF mixture (*n*_QUER_:*n*_CAFF_:*n*_DPPC_ = 1:2:10 ratio → *n*_QUER+CAFF_:*n*_DPPC_ = 3:10 ratio) was almost exactly the same as that of QUER at *n*_QUER_:*n*_DPPC_ = 1:10 ratio. This shows that in the case of the QUER/CAFF mixture, QUER alone is responsible for the shift in *T*_m_, and the effect of CAFF was not detectable. This was somehow surprising, as CAFF previously showed synergistic effects in the case of a gallic acid/CAFF mixture [68]. Pure CAFF, on the other hand, showed negligible effects on DPPC bilayers (Figure 4). However, since the effect of the compound can be membrane-specific, as previously demonstrated in the case of QUER [70], and since the size and shape of liposomes seemingly changes the effects on bilayers as well [23], synergistic effects might be absent only in the case of DPPC lipids. In any case, the observations in Figure 8 are also further supported by the changes in enthalpy shown in Table 3, where the enthalpy of the main thermally induced phase transition of DPPC with the added CAFF/QUER mixture is 28.9 kJ/mol, whereas the enthalpy of the thermally induced main phase transition of DPPC with added QUER in comparable molar ratio (*n*_QUER_:*n*_DPPC_ = 3:10) is 24.3 kJ/mol, and with added CAFF in comparable molar ratio (*n*_CAFF_:*n*_DPPC_ = 3:10), it is 33.8 kJ/mol. Further, we can see from Table 3 that the enthalpies of the thermally induced phase transition of DPPC with CAFF and pure DPPC are the same (Δ*H_1+2_* = 33.8 kJ/mol).

Bliss independence was used to confirm the synergy of the CHRY/QUER compound combination. Unlike dose-dependent drug response, Bliss independence is easier to compute, because it models the combined effect (*E*_total_) as the product of the individual effect with compound A (*E*_A_) and B (*E*_B_):$$E_{\mathrm{total}}=E_{A}\times E_{B}$$
where each effect (*E*) is expressed as the percentage of the control effect [71,72]. We have used the enthalpy change in pure DPPC transition (Δ*H*_1+2_ (DPPC)) as the control effect, as it corresponds to the transition enthalpy of the ˝pure˝ liposomes in our experiments. Adding QUER or CHRY to DPPC reduces transition enthalpy to 85 and 96%, respectively. According to Bliss independence, the predicted reduction in transition enthalpy when both compounds are combined is 82%. The observed reduction in transition enthalpy is less than 59% (Figure 9A, QUER+CHRY column), which is much lower than the predicted effect, thus suggesting that the QUER/CHRY combination is synergetic (Figure 9A). On the other hand, the predicted reduction in transition enthalpy (85%) when adding the QUER/CAFF combination matches the observed reduction in transition enthalpy (86%). This suggests that the QUER/CAFF combination is very likely additive (Figure 9B).

Thus, based on the comparison of the results from Figure 4, Figure 7 and Figure 8, it can be assumed that CAFF and *t*-FER do not contribute to the lowering of the *T*_m_ of the main DPPC lipid phase transition in *Mix*. Initially, this seems to explain why the transition temperature of the *Mix* is similar to pure QUER (at *n*_QUER_:*n*_DPPC_ = 1:2 ratio) (Table 2), since only CHRY, QUER, and EGCG actually contribute to the effect on *T*_m_. However, despite assuming that CAFF and *t*-FER have no effect on *T*_m_, the combined molar ratio of CHRY, QUER, and EGCG in the mixture is still 5:10 (*n*_CHRY_:*n_QUER_*:*n*_EGCG_:*n*_DPPC_ = 1:2:1:10 → *n*_CHRY+QUER+EGCG_:*n*_DPPC_ = 5:10), which is the same as the ratio of pure QUER (*n*_QUER_:*n*_DPPC_ = 1:2) in a sample in which pure QUER was added to DPPC liposomes. This is somehow surprising, since in the case of the mixture of QUER and CHRY (Figure 7), the synergistic effect between the two compounds was clearly seen, although the ratio of the mixture was the same as the molar ratio of pure QUER (*n*_QUER_:*n*_DPPC_ = 3:10). This discrepancy could be interpreted in two ways: i) one or more of the five compounds in the mixture had an antagonistic effect on at least one of the CHRY, QUER, or EGCG; ii) synergism between EGCG, QUER, and CHRY is present, but pure QUER is still more potent at half the molar ratio chosen (*n*_QUER_:*n*_DPPC_ = 1:2). Since it has been demonstrated that QUER and CHRY show some kind of synergism (Figure 7), that CAFF does not show any kind of antagonism in the case of the QUER/CAFF mixture (Figure 8), and that the influence of EGCG on the *T*_m_ of the bilayer is stronger than that of pure CHRY, the reason for the effect of the *Mix* is most likely the complex interactions between all five compounds. However, a possible antagonism cannot be excluded beyond doubt, since in the case of certain other molecules, e.g., a CAFF/naringenin mixture, both antagonistic and synergistic effects were observed depending on the bacterial strain tested [36]. Therefore, based on all available data, it can be concluded that it is very difficult to predict the synergistic mechanism(s) of compound mixtures, and that the synergism of compounds would have to be investigated individually for each mixture, especially if different model membranes/bacterial strains/cells are being used. This type of conclusion is also supported by the fact that each propolis sample (with unique/different composition) usually has a very different effect on selected microorganisms, cells, antioxidant activities, and many other properties [2,57,73].

The possible synergism of compounds was additionally checked with the TBARS assay. In this case, the *Mix* was prepared at identical *w*_compound(s)_:*w*_PL-90G_ mass ratios (*w*_CHRY_:*w*_CAFF_:*w_t_*_-FER_:*w*_QUER_:*w*_EGCG_:*w*_PL-90G_ = 1:1:1:1:1:100; final mass ratio of *w_Mix_*:*w*_PL-90G_ = 1:20) and was compared to pure EGCG liposomes (*w*_EGCG_:*w*_PL-90G_ = 1:20) in order to observe any difference between the antioxidation potential of the pure compound (EGCG) and the *Mix*. EGCG was chosen since its antioxidant activity is well documented, as previously shown in our laboratory [60]. The results showed that the peroxidation of unsaturated lipids in the form of liposomes throughout the 24 h of exposure to UV light stayed the same in the case of *Mix*, whereas in the case of EGCG, statistically significant peroxidation of lipids occurred after 24 h (Figure 10). As the peroxidation values for all EGCG measurements, except the last one at 24 h, were statistically a bit lower than those of the *Mix*, the rise in peroxidation values after 24 h for EGCG is even more significant. These results additionally confirm our previous thesis, that the mixture of CHRY, CAFF, *t*-FER, QUER, and EGCG exhibit synergistic effects.

Last, but not least, the synergistic effect of compounds found in propolis on the lipid bilayers was additionally tested by comparing the individual compounds to the PP. Since propolis includes the mix of many compounds, we expected that its effect on the membrane will be somehow similar to the effect of the *Mix*, or even more pronounced. The results shown in Figure 11 confirm our hypothesis, as the PP indeed had a pronounced impact on the DPPC lipid model membrane. Moreover, its effect is similar to/stronger than the effect of pure QUER. This is not surprising, as each propolis sample usually includes 80–100 compounds [33], many of which have strong biological activity [5]. However, since not all of the compounds found in propolis have a strong effect on the phase transition of DPPC bilayers (e.g., CAFF and *t*-FER, as shown in this study), at least some degree of synergism among certain compounds must exist in order for propolis to display such an effect. The obtained results definitely confirm this, since at least some degree of synergy between certain compounds in propolis was observed (e.g., QUER and CHRY, as shown in this study), which additionally explains its strong biological activity.

## 3. Materials and Methods

### 3.1. Materials

The 70% (*w*/*w*) poplar propolis ethanol extract (which contained waxes and some other impurities) in 96% EtOH (PP) was kindly provided by Medex d.o.o. (Ljubljana, Slovenia). Encapsulated propolis powder (LIO) was prepared using gum arabic, which was kindly provided by WILD GmbH & Co. KG (Heidelberg-Eppelheim, Germany), as previously described [10]. 1,2-Dipalmitoyl-*sn*-glycero-3-phosphatidylcholine (DPPC) was purchased from Avanti Polar Lipids (Alabaster, AL, USA), phospholipon PL-90G (PL-90G) from Lipoid GmbH (Ludwigshafen, Germany) (94.0–102.0% phosphatidylcholine, max 4.0% lysophosphatidylcholine, max 0.3% tocopherol), 4-(2-hydroxyethyl)-1-piperazineethanesulfonic acid (HEPES), quercetin (QUER), chrysin (CHRY), (-)-epigallocatechin-3-gallate (EGCG), *trans*-ferulic acid (*t*-FER), caffeic acid (CAFF), 2-propanol, triammonium citrate, Tween 80, meat extract, starch, and agar were obtained from Sigma-Aldrich (Hamburg, Germany; Buchs, Switzerland; St. Louis, MO, USA), dimethyl sulfoxide (DMSO), formic acid, methanol (MeOH), chloroform (CHCl_3_), 2-tiobarbituric acid, perchloric acid, sulfuric acid, hydrochloric acid, Dglucose, and trichloroacetic acid were purchased from Merck (Darmstadt, Germany; Molsheim, France), Bacto™ Yeast extract and BBL™ Trypticase™ peptone were obtained from Becton, Dickinson and Company (Le Pont-de-Claix, France), CH_3_COONa × 3 H_2_O, MnSO_4_ × 1 H_2_O and BaCl_2_ × 1 H_2_O from Kemika (Zagreb, Croatia), K_2_HPO_4_, MgSO_4_ × 7 H_2_O and 96% EtOH from Honeywell, Riedel-de Haën (; Seelze, Germany). Water was purified using an Elix Advantage water purification system (EMD Millipore, Darmstadt, Germany) for double distilled water and a Milli-Q Gradient water purification system (EMD Millipore, Darmstadt, Germany) for milli-Q water.

### 3.2. Bacterial Strains and Culture Preparations

In this study, one gram-negative and four gram-positive bacterial strains were used. The representative of gram-negative bacteria was a strain of *Escherichia coli* O157:H7tox^-^ (IM219), and the gram-positive strains were *Lacticaseibacillus rhamnosus* GG (IM239), *Lactobacillus paragasseri* K7 (IM105), *Lactobacillus delbrueckii* subsp. *bulgaricus* Selur6 (IM411), and *Lactiplantibacillus plantarum* LJ-130/1R (IM943). The taxonomy follows [74]. All five bacterial strains were kindly supplied by the Institute of Dairy Science and Probiotics (Domžale, Slovenia) and were cryopreserved at −80 °C. For each experiment, the bacteria were inoculated either in Mueller–Hinton broth or agar (MHB, MHA), in the case of *E. coli*, or in the De Man, Rogosa, and Sharpe (MRS) broth or agar, in the case of gram-positive strains. MHB/MHA were prepared as follows: 2 g of meat extract, 17.5 g of casein hydrolysate (peptone), 1.5 g of starch, 1 L of distilled water, and in the case of MHA, 17 g of agar. The final pH was 7.05 ± 0.10. The MRS broth and agar were prepared as follows: 10 g of casein hydrolysate (peptone), 10 g of meat extract, 4 g of yeast extract, 20 g of D-glucose, 5 g of CH_3_COONa × 3 H_2_O, 2 g of K_2_HPO_4_, 2 g of triammonium citrate, 1 mL of Tween 80, 0.2 g of MgSO_4_ × 7 H_2_O, 0.05 g of MnSO_4_ × 1 H_2_O, 1 L of distilled water, and in the case of MRS agar, 12.4 g of agar. The pH was adjusted to 6.20 ± 0.10 with HCl, and the final pH after autoclaving was 5.85 ± 0.10. After sterilization, 15-20 mL of either MHA or MRS agar were poured into petri-dishes (100 ×15 mm), and after cooling down, both agar plates and broths were stored at 4 °C. The cultures for disc-diffusion agar assays were prepared as follows: cryo-stored cultures were unfrozen, and 50 µL of culture in glycerol was added to 9.5–10 mL of either MHB or MRS broth. The cultures were incubated for 24 h at 37 °C with no additional shaking, except for *E. coli*, which was shaken at 220 rpm. After 24 h, 50 µL of cultures were taken and added to a fresh set of broths (9.5–10 mL) and incubated for 18 h. Again, only *E. coli* broth was shaken at 220 rpm. After the second incubation, solutions with 10^7^ cells/mL were prepared. The bacterial concentrations were determined spectrophotometrically (600 nm), with the help of McFarland standards [75].

### 3.3. Preparation of Liposomes

The liposomes were prepared by the thin film method [76] for obtaining DPPC and PL-90G multi-lamellar vesicles (MLVs) or by the proliposome method [77] for obtaining the PL-90G MLVs.

In the case of DPPC thin film liposomes, approximately 20 mg of DPPC was weighted into the 10 mL rotary flask, then the lipids were dissolved in 4 mL of MeOH:CHCl_3_ (3:7, *v*/*v*) mixture. The solvent was then completely evaporated at 180–250 mbar using a rotary vacuum evaporator (Rotavapor R-210, Büchi, Switzerland) while heating the flask in the water bath at 35 °C with the rotation speed set at 3. After formation, the thin film was dried for at least 2.5 h at high vacuum (10–15 mbar) until constant weight. The thin film was then heated to 50 °C in a water bath, and the appropriate volume of preheated (50 °C) 20 mM, pH 7.0 HEPES buffer was added. The suspension was then hydrated at 50 °C in a closed rotary flask for another 45 min with occasional shaking and 30 s sonication in an ultrasonic bath (Bandelin Sonorex TK52; Bandelin Electronic GmbH Q Co. KG, Berlin, Germany). After hydration, the suspension of MLVs was sonicated in an ultrasonic bath for another 3 min before being pipetted into 2 mL centrifuges (300 µL each), aerated with nitrogen gas, frozen in liquid nitrogen, and stored in the freezer at −80 °C until further use. The final concentration of MLVs in the centrifuges was 5 mg/mL.

In the case of PL-90G thin film liposomes, approximately 100 mg of PL-90G was weighted into the 10 mL rotary flask, then the lipids were dissolved in CHCl_3_. In the case of liposomes prepared with the mixture of CHRY, CAFF, *t*-FER, QUER, and EGCG (*Mix*), the first four compounds dissolved in MeOH:CHCl_3_ (3:7, *v*/*v*) mixture were added to the PL-90G lipids to yield *w*_compound_:*w*_PL-90G_ = 1:100 mass ratio. The solvent was then completely evaporated at 150–250 mbar while heating the flask in the water bath at 35 °C with the rotation speed set at 10. After the thin film was formed, it was dried for at least 2.5 h at high vacuum (10–15 mbar) until constant weight. In the case of liposomes prepared with the *Mix* or EGCG alone, the appropriate amount of EGCG was added to the liposomes in a HEPES (20 mM, pH 7.0) solution. In the case of *Mix*, EGCG was added to yield a *w*_EGCG_:*w*_PL-90G_ = 1:100 (*w_Mix_*:*w*_PL-90G_ = 1:20), and in the case of EGCG alone, the final ratio was *w*_EGCG_:*w*_PL-90G_ = 1:20. The thin film was then heated to 50 °C in a water bath, and the appropriate volume of preheated (50 °C) 20 mM, pH 7.0 HEPES buffer was added, to yield a final concentration of 10 mg/mL PL-90G. The suspension was then hydrated at 50 °C in a closed rotary flask for another 45 min with occasional shaking and 30 s sonication in ultrasonic bath. After hydration, the suspension of MLVs was sonicated in an ultrasonic bath for another 3 min before being used. The suspensions of PL-90G (with or without added compounds) were either used immediately or stored for a maximum of 24 h at 2–4 °C after being aerated with nitrogen gas.

For the empty proliposomes, 100 mg of PL-90G was weighted in a small glass beaker, then 127 µL of 96% EtOH and 200 µL of 20 mM, pH 7.0 HEPES buffer were added, to a final ratio of *w*_PL-90G_:*w*_ethanol_:*w*_buffer_ = 1:1:2. The mixture was heated at 60 °C in the water bath while being stirred on a magnetic stirrer at 800 rpm. After the PL-90G was completely dissolved and ethanol evaporated, the glass beaker was cooled to a room temperature, and during stirring (800 rpm), 9.8 mL of HEPES buffer was gradually added in small droplets, to yield a final volume of 10 mL and a final liposomal concentration of 10 mg/mL. The formed MLVs were hydrated and stirred for 1 h. The liposome suspension was then aerated with nitrogen gas and stored in the refrigerator at 4 °C for a maximum of two days. For the proliposomes including PP or LIO (final ratios of *w*_propolis_:*w*_PL-90G_ = 1:10 and/or 1:100), the lipids were either dissolved in 127 µL of appropriately diluted PP in 96% EtOH (instead of in pure 96% EtOH) or hydrated with an appropriately concentrated LIO dispersed in 9.8 mL of HEPES (instead of in pure HEPES).

### 3.4. Differential Scanning Calorimetry

To follow the temperature of gel-to-liquid crystalline phase transition (*T*_m_) of the lipids, different ratios of different compounds or their combinations were added to DPPC liposomes, and the DSC was carried out on a NANO DSC series III (Calorimetry Science Corp., Provo, UT, USA). The compounds added to the liposomes were: caffeic acid, chrysin, quercetin, *trans*-ferulic acid, chrysin/quercetin mixture, caffeic acid/quercetin mixture (all dissolved in DMSO), EGCG, LIO (both dissolved in HEPES), PP (dissolved in 96% EtOH), and a mixture of CHRY, QUER, EGCG, CAFF, and *t*-FER (*Mix)* (dissolved in DMSO). The maximal concentrations of EtOH and DMSO in the suspensions never exceeded 1%. The molar ratios of the added compounds were *n*_compound_:*n*_DPPC_ = 1:1, 1:5, 3:10, and 1:10 for CHRY; 1:1, 1:5, and 1:10 for EGCG; 1:1 and 1:5 for CAFF and *t*-FER; 1:2, 1:5, 3:10, and 1:10 for QUER; 9:10 for *Mix*; and 3:10 for CHRY/QUER and CAFF/QUER mixtures, and the weight ratios for the propolis:DPPC were 1:10 and 1:20, for both PP and LIO. The controls were prepared with the DPPC liposomes in HEPES buffer, HEPES buffer with 1% DMSO or ethanol, or in HEPES buffer with the addition of pure gum Arabic in the same concentration as for the *w*_LIO_:*w*_DPPC_ ratio. The final concentration of the DPPC was 1 mg/mL in all samples. The samples were sonicated in an ultrasonic bath for 3 min, degassed under vacuum for at least 20 min, and loaded into the calorimetric cell, in which they were heated/cooled repeatedly in the temperature range of 20–70 °C, at the heating/cooling rate of 1 °C/min, under increased pressure (additional 3 atmospheric pressures). We subtracted the corresponding baseline (buffer–buffer) scans from the heating/cooling sample scans and divided the resulting heat flow by DPPC mass and heating/cooling rate, thus obtaining the change in heat capacity, Δ*C_p_*, per mol of DPPC as a function of temperature. The observed heat effects were characterized by calculating the change in enthalpy as the area under the experimental curve, and transition temperatures were determined as the curve peak position. An amount of 20 mM, pH 7.0 HEPES buffer was used as the reference. The first scans were used for determining the temperature(s) of pre-transition (*T*_pre_) and the main phase transition(s) (*T*_1_ and *T*_2_), as well as the change in the enthalpy of the main transition (Δ*H*). The subsequent scans were used to assess the reversibility of the lipid phase transition. The NanoAnalyze 3.11.0 and OriginPro 8.1 software were used to evaluate the data from the DSC thermograms.

### 3.5. Antimicrobial Activity of Propolis Extract/Powder Based on Agar Disc-Diffusion Assay

The antimicrobial effects of PP and LIO were determined using the Kirby–Bauer disc-diffusion method [78], with modifications. Briefly: 6 mm discs were cut out of cellulose filter paper (389; Sartorius AG, Göttingen, Germany) and sterilized. Then, 100 µL of bacterial suspensions (10^7^ cells/mL), prepared as described in 2.2., were evenly spread across petri-dishes and left to dry. Then, the sterilized paper discs were evenly placed across each petri dish (7 discs/plate–1 central and 6 around it in hexagonal position), and 10 µL/disc of different PP/LIO/EtOH/water solutions were added. The plates were then left undisturbed for 10 min in order for suspensions to dry. For PP assays, 200 mg propolis/mL (in 100% EtOH), as well as 10, 2, 0.2, and 0.02 mg of propolis/mL (all in 50% EtOH) suspensions were used, and 50 and 100% EtOH were used as negative controls. For LIO assays, 10, 5, 2, 0.2, and 0.02 mg of propolis/mL (in milli-Q) suspensions were used, and 50% EtOH and milli-Q water were used as negative controls. The petri dishes were then incubated at 37 °C for 24 h. *E. coli* was incubated aerobically, and the rest were incubated under anaerobic conditions in an AnaeroPack rectangular jar (Mitsubishi gas chemical company, Inc., Tokyo, Japan) using a GENbox anaer (Biomerieux SA, Marcy-l’Étoile, France) for the creation of an anaerobic atmosphere. After 24 h, the inhibition zones that formed around the discs were measured, including the sizes of the discs (6 mm). All assays were prepared in duplicates.

### 3.6. Thiobarbituric Acid Reacting Substances (TBARS) Assay

The peroxidation of the PL-90G liposomes prepared by the proliposome or thin film method (10 mg/mL) with and without different added compounds (propolis/EGCG/*Mix*) was determined spectrophotometrically using the TBARS assay [79]. In brief: 2–2.4 mL of either pure liposomes (control) or liposomes with either propolis, EGCG, or *Mix* at different *w*_compound_:*w*_PL-90G_ ratios (1:10 for PP, 1:100 for PP and LIO, and 1:20 for EGCG and *Mix*) were pipetted in special 2.5 mL UV cuvettes with caps, which were positioned horizontally with the transparent side facing the UV light. The liposomes were then exposed to the UV radiation at 254 nm, 50 Hz (Universal UV Lamp, Camag, Muttenz, Switzerland) for 24 h. Unexposed control samples were also prepared identically, but were kept in the dark. During the 24 h incubation, small aliquots (0.2 mL) were taken from the cuvettes at various time intervals (0, 2, 3, 6, and 24 h for PP and LIO and 0, 1, 2, 4, 6, and 24 for EGCG and *Mix*). The aliquots were then mixed with 3 mL of 20% trichloroacetic acid and 1 mL of a mixture of 1% thiobarbituric and 10% perchloric acid in the test tube with a plastic screw cap. The suspension was heated for 25 min at 100 °C in a water bath. After heating, the reaction was stopped by cooling the tubes in an ice bath (1–3 °C) for 5 min, followed by centrifugation at 1015 rcf for 8 min. The supernatants were then carefully transferred to plastic PS cuvettes, and the absorbance of the samples was measured at 532 nm using an Eppendorf BioSpectrometer^®^ basic spectrophotometer (Eppendorf AG, Hamburg, Germany). For the blank sample, 0.2 mL of 20 mM, pH 7.0 HEPES buffer was added instead of the liposomes sample. The pink color of the supernatant was from the reaction between the thiobarbituric acid and the lipid hydroperoxide products.

### 3.7. Statistical Analysis

The data are presented as means ± SD (standard deviation). Student’s t-tests were performed to differentiate between the means with a 95% confidence interval (*p* > 0.05). Figure 1 was created using ACD-Chem sketch (ACD/Labs, Toronto, ON, Canada), and Figure 2, Figure 3, Figure 4, Figure 5, Figure 6, Figure 7, Figure 8, Figure 9, Figure 10 and Figure 11 were created using OriginPro 2015 32Bit (origin-Lab, Northampton, MA, USA).

## 4. Conclusions

The comparison between ethanolic propolis extract and its encapsulated form (in gum Arabic) showed that encapsulated propolis exhibited lower in vitro effects on membranes, inhibition of bacteria, and inhibition of UV-induced oxidation. However, it should be noted that encapsulated compounds take longer to be fully released from the wall matrix (in this case, gum Arabic) [10,41]; thus, their full effect on membranes, bacteria, etc. may be somehow delayed, especially in certain matrices (e.g., water solutions). Consequently, the results prove that propolis extract has a faster and more pronounced immediate effect, whereas in the case of encapsulated compounds, the immediate effects are not as pronounced, which could provide benefits for ease of handling and target delivery in some specific applications. The TBARS assay also showed that propolis ethanolic extract can act as a good inhibitor of UV-induced oxidation, and consequently, that propolis could be used in various cremes and lotions to protect the skin, or for prolonging the shelf life of certain products. From the DSC results, we can conclude that certain combinations of compounds in propolis undoubtedly show synergistic effects (CHRY and QUER), whereas others have no positive or negative effect on the lipid bilayer (QUER and CAFF). Furthermore, it has been shown that complex mixtures (e.g., *Mix*) can exhibit behaviors that cannot be easily predicted or explained. Therefore, as we have demonstrated for our liposomal system, each mixture should be studied separately, especially for different membrane systems.

## Figures and Tables

**Figure 1 molecules-28-00712-f001:**
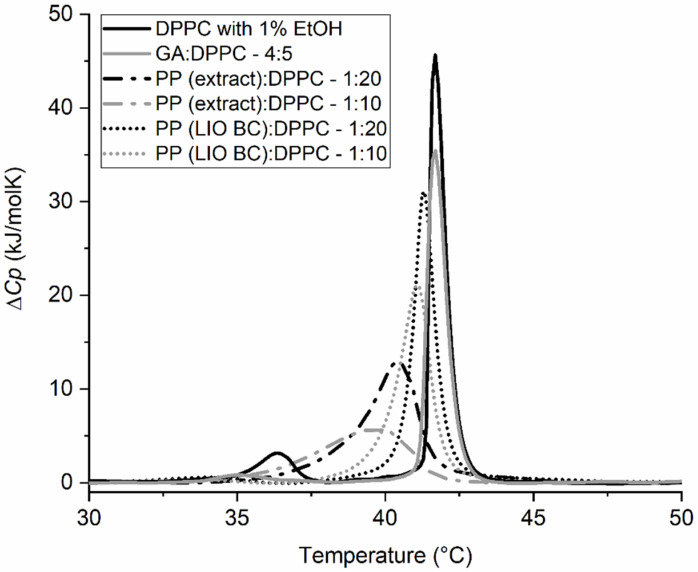
DSC thermograms of 1,2-dipalmitoyl-*sn*-glycero-3-phosphatidylcholine (DPPC) liposomes in 20 mM HEPES (pH = 7.0) buffer with 1% ethanol (EtOH), gum Arabic (GA) (negative controls), and non-encapsulated propolis (PP (extract)) or propolis encapsulated in gum Arabic by lyophilisation (PP (LIO BC)) at different *w*_compound_:*w*_DPPC_ mass ratios. The mass ratio between propolis in gum Arabic inside the encapsulated powders was *w*_propolis_:*w*_gum arabic_ = 1:4.

**Figure 2 molecules-28-00712-f002:**
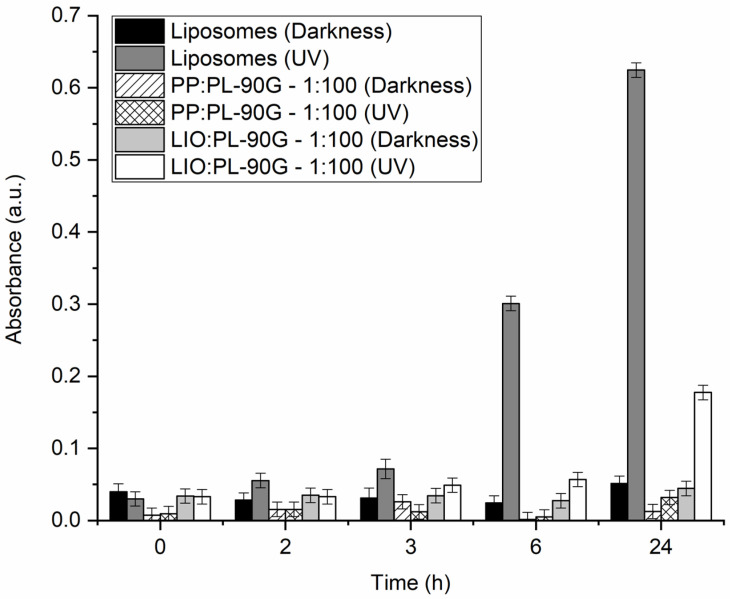
Peroxidation of phospholipon PL-90G (PL-90G) lipids in 20 mM HEPES (pH = 7.0) buffer without (Liposomes) or with the addition of propolis extract (PP) or propolis encapsulated in gum Arabic (LIO) at *w*_PP/PP (LIO)_:*w*_PL-90G_ = 1:100 mass ratio, as obtained by the thiobarbituric acid reactive species (TBARS) assay. Darkness—samples kept in darkness (negative control); UV—samples exposed to ultraviolet light (254 nm).

**Figure 3 molecules-28-00712-f003:**
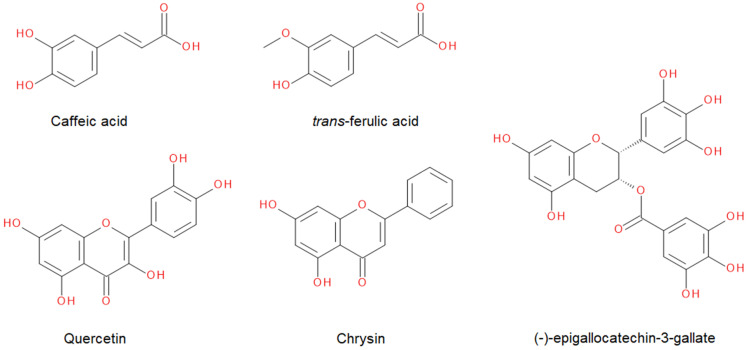
Chemical structures of isolated compounds from different propolis samples.

**Figure 4 molecules-28-00712-f004:**
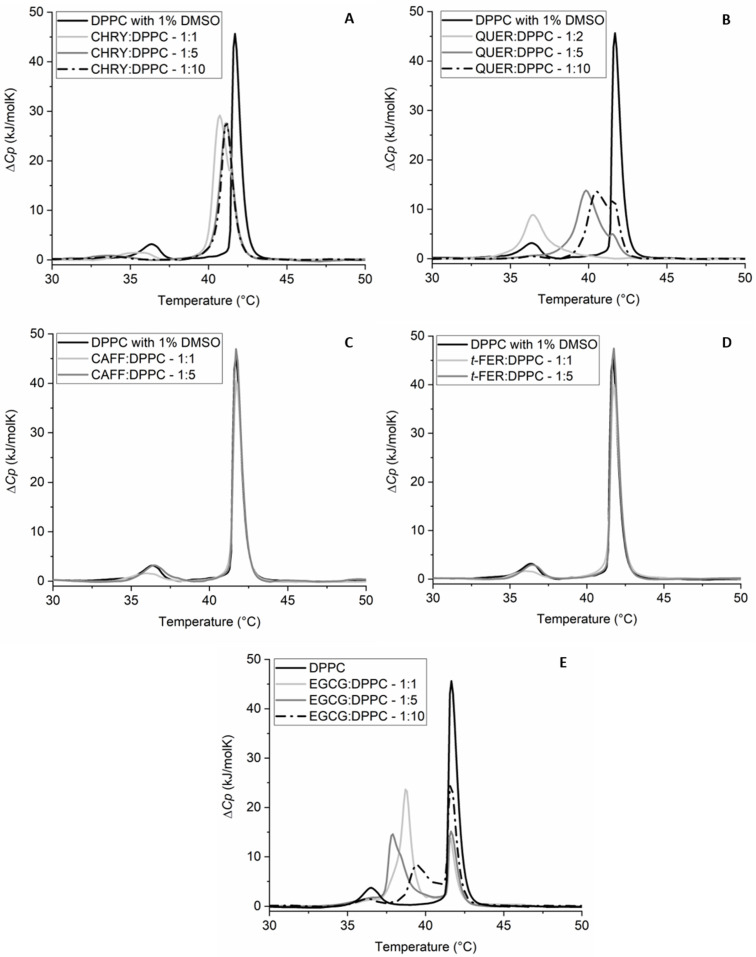
DSC thermograms of 1,2-dipalmitoyl-*sn*-glycero-3-phosphatidylcholine (DPPC) liposomes in 20 mM HEPES (pH = 7.0) buffer with or without the addition of 1% dimethyl sulfoxide (DMSO) (negative controls) and in the presence of chrysin (CHRY) (**A**), quercetin (QUER) (**B**), caffeic acid (CAFF) (**C**), and *trans*-ferulic acid (*t*-FER) (**D**) (all in 1% DMSO and HEPES) or (-)-epigallocatechin-3-gallate (EGCG) (**E**) (in HEPES) at different *n*_compound_:*n*_DPPC_ molar ratios. Δ*C_p_* is the change in heat capacity.

**Figure 5 molecules-28-00712-f005:**
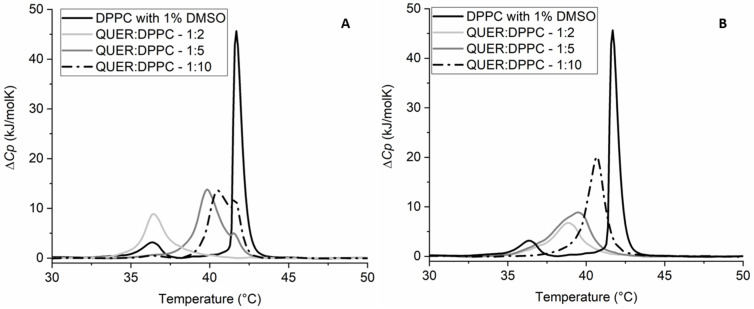
DSC thermograms of 1,2-dipalmitoyl-*sn*-glycero-3-phosphatidylcholine (DPPC) liposomes in 20 mM HEPES (pH = 7.0) buffer with or without the addition of 1% dimethyl sulfoxide (DMSO) (negative controls) and in the presence of quercetin (QUER) at different *n*_compound_:*n*_DPPC_ molar ratios. (**A**)—first heating scan; (**B**)—second heating scan. Δ*C_p_* is the change in heat capacity.

**Figure 6 molecules-28-00712-f006:**
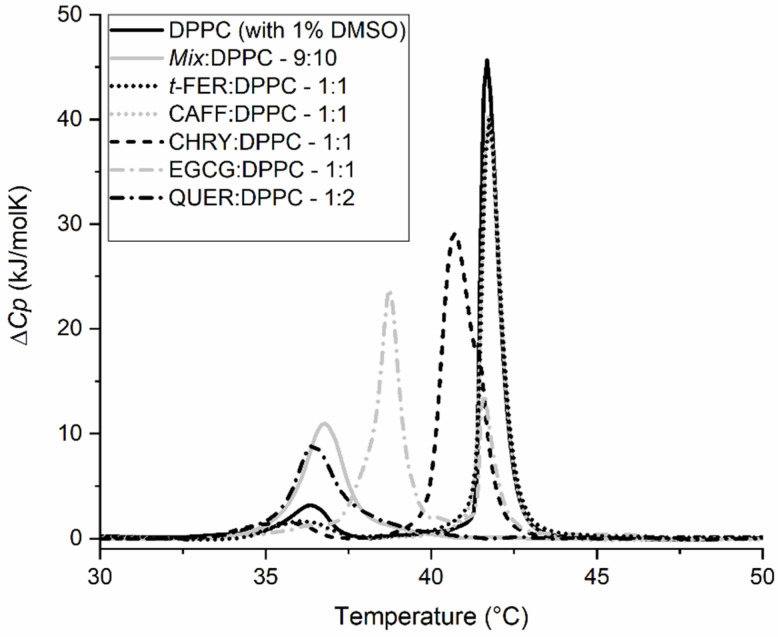
DSC thermograms of 1,2-dipalmitoyl-*sn*-glycero-3-phosphatidylcholine (DPPC) liposomes in 20 mM HEPES (pH = 7.0) buffer and 1% dimethyl sulfoxide (DMSO) (DPPC (with 1% DMSO); negative control), chrysin (CHRY), caffeic acid (CAFF), *trans*-ferulic acid (*t*-FER), quercetin (QUER) (all in 1% DMSO and HEPES), (-)-epigallocatechin-3-gallate (EGCG) (in HEPES), or their mixture (*Mix*) (in 1% DMSO and HEPES) at different *n*_compound/*Mix*_:*n*_DPPC_ molar ratios. The molar ratios of the individual compounds for the *n_Mix_*:*n*_DPPC_ = 9:10 molar ratio was *n*_CHRY_:*n*_CAFF_:*n_t_*_-FER_:*n*_QUER_:*n*_EGCG_:*n*_DPPC_ = 2:2:2:1:2:10. Δ*C_p_* is the change in heat capacity.

**Figure 7 molecules-28-00712-f007:**
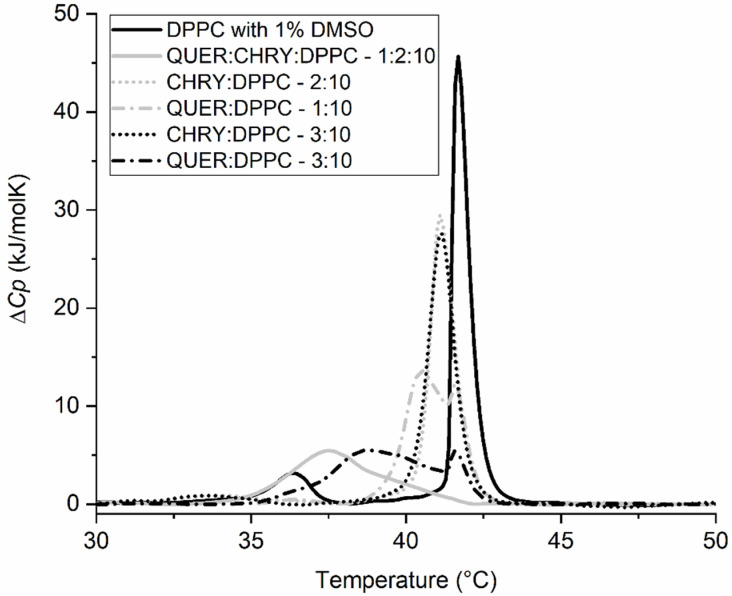
DSC thermograms of 1,2-dipalmitoyl-*sn*-glycero-3-phosphatidylcholine (DPPC) liposomes in 20 mM HEPES (pH = 7.0) buffer and 1% dimethyl sulfoxide (DMSO) (negative control), and in the presence of chrysin (CHRY), quercetin (QUER), or their mixture at different *n*_compound(s)_:*n*_DPPC_ molar ratios. Δ*C_p_* is the change in heat capacity.

**Figure 8 molecules-28-00712-f008:**
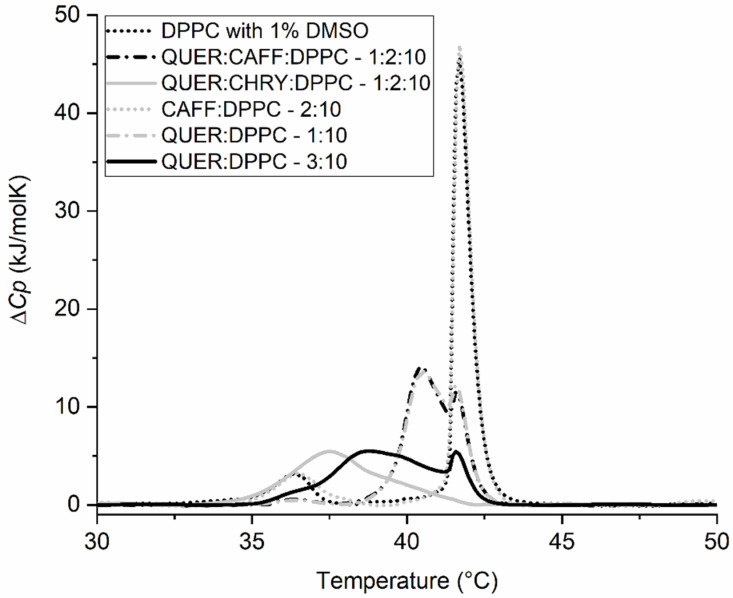
DSC thermograms of 1,2-dipalmitoyl-*sn*-glycero-3-phosphatidylcholine (DPPC) liposomes in 20 mM HEPES (pH = 7.0) buffer and 1% dimethyl sulfoxide (DMSO) (negative control), and the addition of caffeic acid (CAFF), quercetin (QUER), or the mixtures of QUER with either CAFF or chrysin (CHRY) (all in 1% DMSO and HEPES) at different *n*_compound(s)_:*n*_DPPC_ molar ratios. Δ*C_p_* is the change in heat capacity.

**Figure 9 molecules-28-00712-f009:**
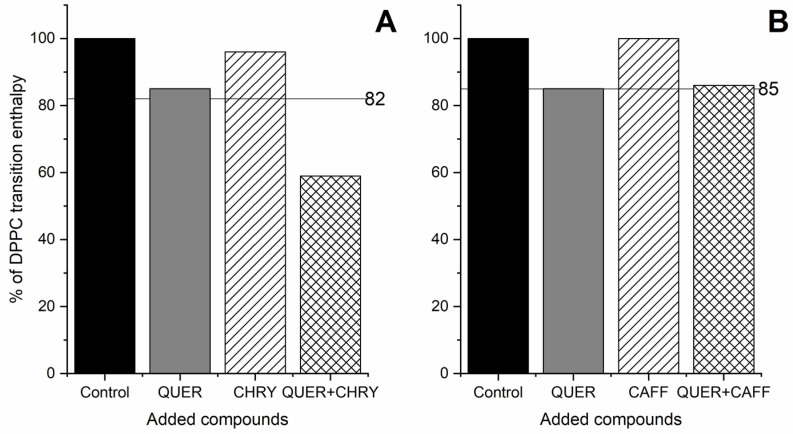
Synergy of compound combination calculated by Bliss independence. (**A**) Predicted reduction in transition enthalpy according to Bliss independence (horizontal line—82%) and observed reduction in transition enthalpy (QUER+CHRY column) when adding QUER/CHRY mixture to DPPC; (**B**) predicted reduction in transition enthalpy according to Bliss independence (horizontal line—85%) and observed reduction in transition enthalpy (QUER+CAFF column) when adding QUER/CAFF mixture to DPPC. QUER—quercetin; CAFF—caffeic acid; CHRY—chrysin; DPPC—1,2-dipalmitoyl-*sn*-glycero-3-phosphatidylcholine. The *n*_compound(s)_:*n*_DPPC_ molar ratios were as follow: QUER:DPPC—1:10; CHRY:DPPC—2:10; CAFF:DPPC—2:10; QUER:CHRY:DPPC—1:2:10; QUER:CAFF:DPPC—1:2:10; control—pure DPPC (with 1% DMSO).

**Figure 10 molecules-28-00712-f010:**
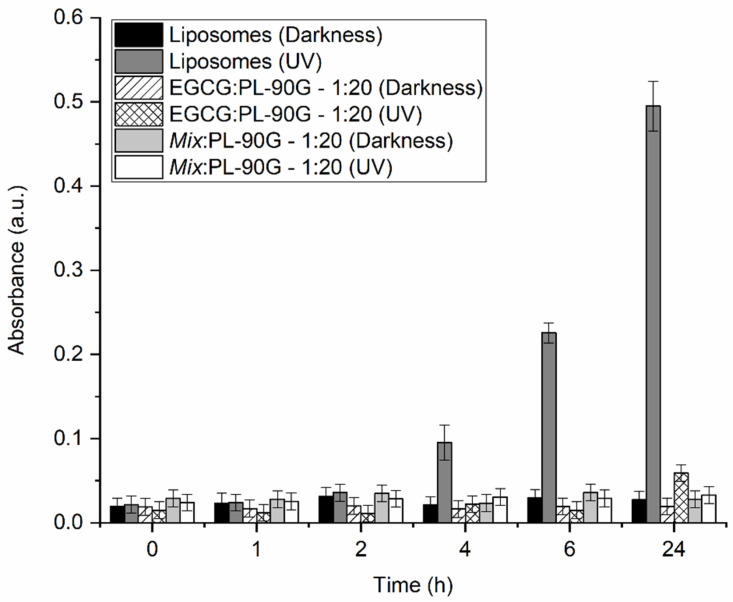
Peroxidation of phospholipon PL-90G (PL-90G) liposomes in 20 mM HEPES (pH = 7.0) buffer without (Liposomes) or with the addition of (-)-epigallocatechin-3-gallate (EGCG) or a mixture of chrysin, caffeic acid, *trans*-ferulic acid, quercetin, and EGCG (*Mix*) at *w*_EGCG/*Mix*_:*w*_PL-90G_ = 1:20 mass ratio, as obtained by the thiobarbituric acid reactive species assay. The mass ratio of compounds in the *Mix* was *w*_chrysin_:*w*_caffeic acid_:*w_trans_*_-ferulic acid_:*w*_quercetin_:*w*_EGCG_ = 1:1:1:1:1. Darkness—samples kept in darkness (negative control); UV—samples exposed to the ultraviolet light (254 nm).

**Figure 11 molecules-28-00712-f011:**
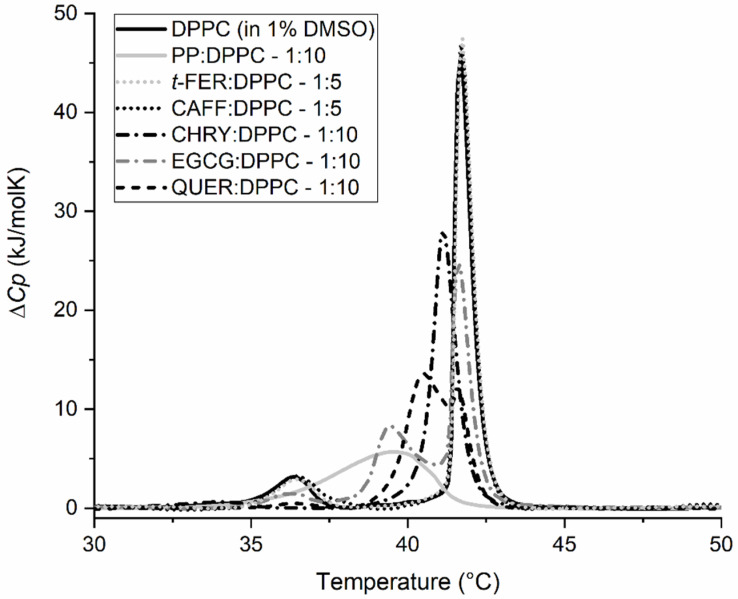
DSC thermograms of 1,2-dipalmitoyl-*sn*-glycero-3-phosphatidylcholine (DPPC) liposomes in 20 mM HEPES (pH = 7.0) buffer and 1% dimethyl sulfoxide (DMSO) (negative control), chrysin (CHRY), caffeic acid (CAFF), *trans*-ferulic acid (*t*-FER), quercetin (QUER) (all in 1% DMSO and HEPES), (-)-epigallocatechin-3-gallate (EGCG) (in HEPES), and propolis extract (PP) (in 1% EtOH and HEPES) at different *n*_compound_:*n*_DPPC_ molar ratios. Δ*C_p_* is the change in heat capacity.

**Table 1 molecules-28-00712-t001:** Antimicrobial activity of solutions containing different amounts of propolis extract (in EtOH) expressed as inhibition zone radiuses, based on the agar disc diffusion test with addition of 10 µL solution/6 mm disc.

Suspensions (mg of Propolis/mL)	*E. coli*	*L. bulgaricus*	*L. paragasseri*	*L. plantarum*	*L. rhamnosus*
100% EtOH	/	11.3 ± 3.0	13.8 ± 0.8	12.8 ± 1.3	14.3 ± 0.4
50% EtOH	/	/	/	/	7.5 ± 0.5 *
Milli-Q water	/	/	/	/	/
200 mg/mL (PP) (100% EtOH)	/	18.5 ± 2.5	16.0 ± 0.5	13.5 ± 0.5	17.0 ± 0.5
10 mg/mL (PP) (50% EtOH)	/	8.5 ± 0.5	7.8 ± 0.8	6.5 ± 0.5 *	9.5 ± 0.5
2 mg/mL (PP) (50% EtOH)	/	/	/	/	8.0 ± 1.0
0.2 mg/mL (PP) (50% EtOH)	/	/	/	/	8.0 ± 0.5 *
0.02 mg/mL (PP) (50% EtOH)	/	/	/	/	7.3 ± 0.8 *

/—no inhibition zone detected (inhibition zones measured 6 mm, which is the size of the disc radius); * only partial inhibition zone detected (smaller and fewer colonies, but without complete inhibition); PP—propolis extract; 6 mm disc radius is included in all measurements; the results for 0.02, 0.2, 2, 5, and 10 mg/mL of LIO (all in water) are not shown, as there was no antibacterial activity of LIO.

**Table 2 molecules-28-00712-t002:** The thermodynamic profile of the 1,2-dipalmitoyl-*sn*-glycero-3-phosphocholine lipids (MLV) at pH 7.0, at different molar ratios (*n:n*) of chrysin, quercetin, caffeic acid, *trans*-ferulic acid, (-)-epigallocatechin-3-gallate, and their mixture.

Compound	Ratio (*n:n*)	*T*_pre_ (°C)	*T*_1_ (°C)	*T*_2_ (°C)	Δ*H*_pre_ (kJ/mol)	Δ*H*_1+2_ (kJ/mol)
CHRY:DPPC	1:1	35.6 ± 0.2	40.7 ± 0.1	/	3.6 ± 0.5	40.1 ± 1.0
CHRY:DPPC	1:5	33.7 ± 0.2	41.2 ± 0.1	/	2.8 ± 0.5	32.6 ± 1.0
CHRY:DPPC	1:10	34.0 ± 0.2	41.2 ± 0.1	/	1.8	30.1 ± 1.0
QUER:DPPC	1:2	/	36.4 ± 0.1	/	/	19.5 ± 1.0
QUER:DPPC	1:5	/	39.8 ± 0.1	41.5 ± 0.1	/	30.6 ± 1.0
QUER:DPPC	1:10	36.4 ± 0.2	40.5 ± 0.1	41.6 ± 0.1	0.7 ± 0.1	28.8 ± 1.0
CAFF:DPPC	1:1	35.9 ± 0.2	41.7 ± 0.1	/	3.4 ± 0.5	31.9 ± 1.0
CAFF:DPPC	1:5	36.5 ± 0.2	41.7 ± 0.1	/	5.7 ± 0.5	33.8 ± 1.0
*t*-FER:DPPC	1:1	36.0 ± 0.2	41.7 ± 0.1	/	3.5 ± 0.5	34.0 ± 1.0
*t*-FER:DPPC	1:5	36.4 ± 0.2	41.7 ± 0.1	/	4.8 ± 0.5	34.8 ± 1.0
EGCG:DPPC	1:1	/	38.7 ± 0.1	41.6 ± 0.1	/	38.4 ± 1.0
EGCG:DPPC	1:5	/	37.9 ± 0.1	41.6 ± 0.1	/	35.9 ± 1.0
EGCG:DPPC	1:10	36.2 ± 0.2	39.4 ± 0.1	41.7 ± 0.1	2.7 ± 0.5	34.1 ± 1.0
*Mix*:DPPC	9:10	/	36.8 ± 0.1	/	/	20.7 ± 1.0
DPPC (1% DMSO)	/	36.4 ± 0.2	41.7 ± 0.1	/	5.4 ± 0.5	33.8 ± 1.0
DPPC	/	36.5 ± 0.2	41.6 ± 0.1	/	5.6 ± 1.0	34.2 ± 1.0

*T*_pre_—transition temperature of the pre-transition peak; *T*_1_—transition temperature of the first peak; *T*_2_—transition temperature of the second peak; Δ*H*_pre_—enthalpy of the pre-transition; Δ*H*_1+2_—enthalpy of the main transition; QUER—quercetin; CAFF—caffeic acid; CHRY—chrysin; *t*-FER—*trans*-ferulic acid; EGCG—(-)-epigallocatechin-3-gallate; DPPC—1,2-dipalmitoyl-*sn*-glycero-3-phosphatidylcholine. The error in enthalpy was estimated during the baseline removal process, and the error in temperature was estimated based on the repetition results.

**Table 3 molecules-28-00712-t003:** The thermodynamic profile of the 1,2-dipalmitoyl-*sn*-glycero-3-phosphocholine lipids (MLV) at pH 7.0, at different molar ratios (*n:n*) of chrysin, quercetin, caffeic acid, and their binary mixtures.

Compound	Ratio (*n:n*)	*T*_pre_ (°C)	*T*_1_ (°C)	*T*_2_ (°C)	Δ*H*_pre_ (kJ/mol)	Δ*H*_1+2_ (kJ/mol)
QUER:CAFF:DPPC	1:2:10	36.4 ± 0.2	40.4 ± 0.1	41.6 ± 0.1	0.8 ± 0.1	28.9 ± 1.0
QUER:CHRY:DPPC	1:2:10	/	37.5 ± 0.1	/	/	19.8 ± 1.0
QUER:DPPC	1:10	36.4 ± 0.2	40.5 ± 0.1	41.6 ± 0.1	0.7 ± 0.1	28.8 ± 1.0
QUER:DPPC	3:10	/	38.8 ± 0.1	41.6 ± 0.1	/	24.3 ± 1.0
CAFF:DPPC	2:10	36.5 ± 0.2	41.7 ± 0.1	/	5.7 ± 0.5	33.8 ± 1.0
CHRY:DPPC	2:10	33.7 ± 0.2	41.2 ± 0.1	/	2.8 ± 0.5	32.6 ± 1.0
CHRY:DPPC	3:10	34.1 ± 0.2	41.1 ± 0.1	/	3.1 ± 0.5	31.5 ± 1.0
DPPC (1% DMSO)	/	36.4 ± 0.2	41.7 ± 0.1	/	5.4 ± 0.5	33.8 ± 1.0

*T*_pre_—transition temperature of the pre-transition peak; *T*_1_—transition temperature of the first peak; *T*_2_—transition temperature of the second peak; Δ*H*_pre_—enthalpy of the pre-transition; Δ*H*_1+2_—enthalpy of the main transition; QUER—quercetin; CAFF—caffeic acid; CHRY—chrysin; DPPC—1,2-dipalmitoyl-*sn*-glycero-3-phosphatidylcholine. The enthalpy errors were estimated during the baseline removal process, and the temperature errors were estimated based on the repetition results.

## Data Availability

The data is available upon request to the corresponding author.

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
