# Peer review of "Comparing the Effects of Encapsulated and Non-Encapsulated Propolis Extracts on Model Lipid Membranes and Lactic Bacteria, with Emphasis on the Synergistic Effects of Its Various Compounds"

_molecules, 2023, doi:10.3390/molecules28020712_

Round 1

Reviewer 1 Report

Manuscript title: Comparing the effects of encapsulated and non-encapsulated 2 propolis extracts on the model lipid membranes and lactic bac- 3 teria, with the emphasis on the synergistic effects of its various 4 compounds

The manuscript has some few minor issues, in addition to one major issue. After handling these few issues, the manuscript will be eligible for publication in Molecules.

Minor issues: 1- Line2 142-145 and lines 173-176 need to be reworded to improve clarity. 

2- in the text of Figure 2 description, you use PP (LIO) while in the manuscript text you use (LIO). Please be consistent.

3- In lines 235 and 240: please replace 'mayor' with 'major'.

Major issue:

The method used for detecting synergism is not shown. A common method to indicate the presence of synergism is the calculation of combination index (CI). Please use a mathematical method to indicate synergism, or cite a reference to confirm that your methodology may still indicate synergism.

Author Response

We would like to thank the reviewer 1 for comments which help us to improve the manuscript.

Reviewer #1: Manuscript title: Comparing the effects of encapsulated and non-encapsulated propolis extracts on the model lipid membranes and lactic bacteria, with the emphasis on the synergistic effects of its various compounds. The manuscript has some few minor issues, in addition to one major issue. After handling these few issues, the manuscript will be eligible for publication in Molecules.

Answer to the reviewer: The authors would like to thank the reviewer for his/her time invested in this manuscript, for pointing out the shortcomings of the manuscript and for his/her suggestions as how to improve it.

Minor issues:

Comment #1: Line 142-145 and lines 173-176 need to be reworded to improve clarity.

Answer #1: The lines 142-145 and 173-176 were improved for clarity.

Comment #2: In the text of Figure 2 description, you use PP (LIO) while in the manuscript text you use (LIO). Please be consistent.

Answer #2: Thank you for pointing out this mistake, which actually stemmed from the mistake made in the figure 2. Previous figure 2 was thus replaced with its corrected version and the accompanying text was corrected as well (PP(LIO) was corrected to LIO).

Comment #3: In lines 235 and 240: please replace 'mayor' with 'major'.

Answer #3: In lines 235 and 240 mayor was replaced with major.

Major issue:

Comment #1: The method used for detecting synergism is not shown. A common method to indicate the presence of synergism is the calculation of combination index (CI). Please use a mathematical method to indicate synergism, or cite a reference to confirm that your methodology may still indicate synergism.

Answer #1: Thank you for pointing out that no method used for detecting synergism is shown in our manuscript. As the reviewer suggested we have used Bliss independence to confirm the synergy of CHRY/QUER compound combination. The results are shown in new Figure 9.

Reviewer 2 Report

The article entitled ‘Comparing the effects of encapsulated and non-encapsulated propolis extracts on the model lipid membranes and lactic bacteria, with the emphasis on the synergistic effects of its various compounds’ is an interesting paper about propolis (encapsulated and free propolis extracts) and the propolis main compounds synergistic effect. 

In general, the work performed is of good quality and I suggest being accepted after minor revision.

Below, there is a list of few suggestions that in my opinion would help to improve the manuscript.

1. I suggest using EtOH, MeOH and CHCl3 instead ‘ethanol, methanol and chloroform’, after first time appear in the manuscript

2. All Figures are clear and appropriate, but please refer Figure 3(line 213) in the text at the appropriate place where it have been first discussed  

3. Please change the Caption of the Figure 3 into ‘Chemical structures of isolated compounds from different propolis samples’ instead ‘Compounds isolated from different propolis samples’-line 213

4. Please check English language spelling grammar of manuscript 

Author Response

ANSWERS TO REVIEWER 2

We would like to thank the reviewer 2 for comments which help us to improve the manuscript.

Reviewer #2: The article entitled ‘Comparing the effects of encapsulated and non-encapsulated propolis extracts on the model lipid membranes and lactic bacteria, with the emphasis on the synergistic effects of its various compounds’ is an interesting paper about propolis (encapsulated and free propolis extracts) and the propolis main compounds synergistic effect. In general, the work performed is of good quality and I suggest being accepted after minor revision.

Answer to the reviewer: The authors would like to thank the reviewer for his/her time invested in this manuscript, for endorsing our work and for pointing out a few details which could be improved.

Minor issues:

Comment #1: I suggest using EtOH, MeOH and CHCl3 instead ‘ethanol, methanol and chloroform’, after first time appear in the manuscript.

Answer #1: EtOH, MeOH and CHCl3 were used in the manuscript instead of ‘ethanol, methanol and chloroform’, after the first time they appeared in the manuscript. For better clarity, however, the word ˝ethanolic propolis extract˝ was not changed to ˝EtOH propolis extract˝ but was always written in full or shortened with PP.

Comment #2: All Figures are clear and appropriate, but please refer Figure 3(line 213) in the text at the appropriate place where it have been first discussed. 

Answer #2: The figure 3 is now referred in the text at the appropriate place where it has been first discussed (see line 191).

Comment #3: Please change the Caption of the Figure 3 into ‘Chemical structures of isolated compounds from different propolis samples’ instead ‘Compounds isolated from different propolis samples’-line 213

Answer #3: The caption of the Figure 3 was changed to the one proposed by the reviewer.

Comment #4: Please check English language spelling grammar of manuscript

Answer #4: The English language spelling grammar of the manuscript was checked as proposed by the reviewer. Some minor mistakes were indeed found and were corrected accordingly. We sincerely hope that the improved manuscript will be sufficient. 
